


# The Record-Breaking Precipitation Event of December 2022 in Portugal

Tiago M. Ferreira[1,2], Ricardo M. Trigo[1], Tomás H. Gaspar[1], Joaquim G. Pinto[2], Alexandre M. Ramos[2]

[1] Instituto Dom Luiz, Faculdade de Ciências, Universidade de Lisboa, Portugal

5 [2] Institute of Meteorology and Climate Research Troposphere Research (IMKTRO), Karlsruhe Institute of Technology (KIT), Karlsruhe, Germany

*Correspondence to*: Tiago M. Ferreira (tiago.ferreira@partner.kit.edu)

**Abstract.** Extreme precipitation events (EPEs) present potentially an enormous societal risk and often lead to major human and economic impacts. In the mid-latitudes, such EPEs are often triggered by intense extratropical cyclones and their associated frontal systems. Over the last decade, several studies have shown the important and specific role played by 10 Atmospheric Rivers (ARs) in the occurrence of EPEs in western Europe, particularly in the Iberia Peninsula. In this study we analyze the all-time 24h record-breaking precipitation (120.3mm) recorded in the historical Dom Luiz Observatory (since 1863) in Lisbon, Portugal, between 12 and 13 December 2022. A synoptic evaluation of surface and upper-level fields from 5 to 14 December is performed using ERA-5 reanalysis. The week before the EPE, there was a combined effect of a large-scale SLP gradient resembling the NAO negative phase, a southerly position of the jet stream and an above normal positive 15 SST anomalies over the North Atlantic leading to the development of several low-pressure systems at relatively low latitudes, all travelling along the same mean path towards western Europe. The atmospheric river associated with this event was first detected on late 10 December associated with a deep extratropical cyclone. The combination of high Integrated Vapor Transport (and moisture inflow by the warm conveyor belt), with a dynamical component characterized by a suitable uplift motion, allowed the system to evolve and maintain its AR characteristics for 72 h. The extratropical cyclone and 20 associated AR moved northeast towards Iberia, making landfall in Portugal, on 12 December, as an extreme AR event, leading to the 24h precipitation record breaking event.

## 1 Introduction

In western Iberia, extreme precipitation events (EPEs) occur primarily in the winter half-year and can lead to landslides 25 and flooding that thus induce major socioeconomic impacts (e.g., Liberato et al., 2012; Trigo et al., 2016). These events have been widely studied in the last decade, along the western Europe (Ramos et al., 2015; Stohl et al., 2008; Lavers et al., 2011). On the synoptic scale, most of these EPEs are associated with the passage of intense extratropical storms, their associated frontal systems and sometimes with the so-called Atmospheric Rivers (Dettinger et al., 2015). Extratropical cyclones (ETCs) are midlatitude synoptic low-pressure systems that develop primarily over the large ocean basins thanks to a range of 30 physical processes, primarily baroclinicity and latent heat release (e.g., Pinto et al., 2009; Schultz et al., 2019). These


systems are largely responsible for the poleward transport of heat and moisture in the climate system (Peixoto and Oort, 1992) and have a major impact on regional temperature, precipitation and cloudiness (e.g., Colle et al. 2015; Hawcroft et al 2018). They are frequently responsible for extreme weather conditions, such as precipitation and wind extremes (Fink et al., 2009), which cause wind damage and flooding either in coastal or inland areas, playing an important role on the hydrological cycle.

According to the AMS Glossary of Meteorology (Ralph et al., 2018), an Atmospheric River (AR) corresponds to a 'long, narrow, and transient corridor of strong horizontal water vapor transport that is typically associated with a low-level jet ahead of the cold front of an ETC'. They are usually identified using a threshold of vertically integrated vapor transport and are typically located in regions where their dynamical and thermodynamical components are relatively large throughout the depth of the lower troposphere (Dacre et al., 2019). The above mentioned "low-level jet ahead of the cold front" is typically associated with the inflow of the warm conveyor belt (WCB) of an ETC (e.g. Dacre et al., 2015; 2019). Their moisture contents can either be from tropical origin (in which they are sometimes connected to Tropical Moisture Exports, e.g. Knippertz et al., 2013), or result from local convergence along its path, the latter being associated sometimes with the presence of an anticyclone on its southeast side that enhances evaporation (Guo et al., 2020; Ralph et al., 2018).

The large amounts of moisture embedded in ARs may lead to EPEs may support the development of intense convective storms in the maritime, (sub-)tropical airmasses that often lead to extreme precipitation and flooding when interacting with the mountainous western coasts of North America and Europe (Dacre et al., 2015; Lee et al., 2022; Guo et al., 2020;). However, it is important to stress that not all ARs are associated with detrimental impacts. In fact, ARs impacts can be, in some cases, beneficial by providing water and, in some cases, end drought conditions (e.g., Dettinger 2013). Furthermore, on the western coast of the Iberian Peninsula, the dynamical component is also an important factor due to the interaction between the ARs and extratropical cyclones, in particular its WCBs. This branch of an ETC consists of an uplift motion, in which the warm and moist air typically ascends by at least 600 hPa in 48h from the boundary layer to the upper troposphere, ahead of the cyclone's cold front (Blanchard et al., 2020; Martínez-Alvarado et al., 2014). In general, the horizontal component of the trajectories is almost parallel to the cold front, either running on the warm side of the surface cold front or riding up the frontal surface. This is the primary cloud and precipitation producing flow within an ETC as it originates in the cyclone's warm sector, resulting on the transport of both latent and sensible heat. Additionally, the outflow of the WCB can split into two flows, one branch that turns cyclonically around the poleward side of the cyclone center, while the other branch turns anticyclonically, and potentially leading to modifications in the cyclone intensity for the former, and the upper-level large-scale flow for the latter (Catto et al., 2015; Martínez-Alvarado et al., 2014; Joos et al., 2023).

The coasts of Western Europe are exposed to the westerly flow, whose intensity is largely controlled by the pressure gradient between the sub-tropical (Azores) high and the sup-polar (Icelandic) low. This corresponds to the canonical North Atlantic Oscillation pattern (NAO; Wanner et al., 2001), which has variability on a multiple range of time scales (Pinto and Raible, 2012). Embedded in the westerly flow are ETCs and sometimes ARs, which contribute to the advection of high moisture contents towards Western Europe. Every year the western coast of Portugal is struck by ARs (e.g., Ramos et al.,



2015; Eiras-Barca et al., 2018). At the Dom Luiz Observatory (IDL) in Lisbon, continuous rainfall measurements exist since December 1863. This makes it the only station with a near 160-year-long record of daily data in Portugal. This station is particularly important to define the maximum registered precipitation in the area and to infer the evolution of rainfall affecting western Iberia. The top three maxima precipitation measured at IDL from 1863 to 2013 are (1) 18 February 2008 with a precipitation value of 118.4 mm (Fragoso et al., 2010); (2) 5 December 1876 with a value of 110.7 mm (Trigo et al.,

2014); and (3) 30 January 2004 with a value of 101.2 mm. Another intense precipitation event occurred in Lisbon on 24 October 2013, with a precipitation value of 102 mm in 24h.

In this work, we focus on the events in December 2022, in particular on the EPE that took place on 12-13 December and strongly affected the Lisbon area. This event led to a 24h precipitation value of 134.6 mm, measured between 12 December 15 UTC, and 13 December 15 UTC (Dias, 2022). According to IPMA (2022), a value of 120.3 mm was measured at IDL

between 12 December 09 UTC, and 13 December 09 UTC, surpassing the previous highest value on 18 February 2008[1]. This event triggered intense flooding in several areas of the Lisbon district, that led the Portuguese Civil Authority to report over 150 incidents (Davies, 2023). The human impacts were likely minimized by the issuing of a red alert by the IPMA, still several people had to be rescued or evacuated from the affected areas. Moreover, the intense precipitation was also registered on the Spanish region of Extremadura and Madrid, flood damage was also reported and over 200 incidents were responded

by the Emergency Centre (Davies, 2023). According to AEMET (Spanish Meteorological Agency), the station at Navalvillar de Ibor in Cáceres Province of Extremadura recorded 120.4 mm of precipitation in 24 hours on 13 December (Davies, 2023).

In this work we aim to provide a comprehensive synoptic characterization of the atmospheric circulation observed prior and during this EPE (between 5 and 14 December), as well as the associated atmospheric river. Specifically, the aim of this study is threefold: (1) to analyze of the spatial distribution of precipitation measurements from 114 Portuguese

meteorological stations; (2) to characterize of the AR life cycle and associated mechanisms leading to the extreme precipitation values; and (3) analyze the pre-conditions associated with the cyclone and AR formation and their development. Section 2 provides a brief description of the data and methodology used to detect ARs and analyze their atmospheric environment. Section 3 is devoted to the description of the spatial distribution of precipitation as well as the new maximum registered at the IDL station. Section 4 describes the large-scale atmospheric conditions throughout the North

Atlantic, from 5 to 11 December. Section 5 presents the analysis of the AR's life cycle, between the first and last time-steps in which the detection algorithm defined it as an AR and the role played by the large-scale atmospheric circulation and the cyclone. Finally, a summary and conclusions are given in section 6.

---

[1] Daily precipitation totals at Portuguese stations are calculated between 09 UTC of the previous day and 09 UTC of current day



## 2 Data and Methodology

### 2.1 Meteorological Data

The large-scale data used on this study is the ECMWF (European Centre for Medium-Range Weather Forecasts) ERA-5 reanalysis (Hersbach et al., 2020). Several fields were extracted from both surface and at specific atmospheric pressure levels. The former include the mean sea-level-pressure (SLP) and the sea surface temperature (SST). The later include: (1) the zonal and meridional components of the vertically integrated water vapor transport (taking into account the entire atmospheric column); (2) 850/500 hPa geopotential height; (3) 200/850 hPa divergence; (4) 500 hPa temperature; (5) 250

100   hPa wind; and (6) total column water vapor. All these fields were obtained with a 0.25º latitude-longitude spatial resolution and a 6-hourly temporal resolution. The vertically integrated water vapor transport (IVT) and the integrated water vapor (IWV) corresponds to the following metrics:

$$IVT = \sqrt{u_q{}^2 + v_q{}^2} , \tag{1}$$

with

$$u_q = IVT_u = \frac{1}{g} \int_{ps}^{p} uq\,dp \qquad (2) \qquad\qquad \text{and} \qquad\qquad v_q = IVT_v = \frac{1}{g} \int_{ps}^{p} vq\,dp \qquad (3)$$

and

$$IWV = \frac{1}{g} \int_{ps}^{p} q\,dp \tag{4}$$

where g is the gravitational acceleration, q is the specific humidity, ps is surface pressure, p is maximum pressure, and u and v are the zonal and meridional wind components (Lavers et al., 2012). This field is computed from the surface to the top of

the atmosphere (1 hPa), considering all 37 pressure levels available in ERA-5. While it is widely known that the upper levels of the atmosphere hold very little humidity, the use of the entire atmospheric column is mostly due to computational convenience. Instead of an additional computation of both IVT and IWV, the ERA-5 reanalysis provides the variables 'total column water vapor' (IWV), 'vertical integral of eastward water vapor flux' ($IVT_u$) and 'vertical integral of northward water vapor flux' ($IVT_v$) as a vertically integrated field considering all available atmospheric levels.

Data on the inflow, outflow, and ascent of moisture within the warm conveyor belt was provided by KIT (Karlsruhe Institute of Technology). The method, developed by Quinting and Grams (2022), uses convolutional neural networks to predict the footprints of the WCB inflow, ascent, and outflow stages by obtaining conditional probabilities of WCB occurrences, from predictors derived from temperature, geopotential height, specific humidity, and horizontal wind components. The WCB air masses originate from the boundary layer in the warm sector of ETCs (WCB inflow, below 800

120   hPa), ascend across the cyclones' warm front (WCB ascent, at least 600 hPa ascent in 48h), and reach the upper troposphere (WCB outflow, above 400 hPa).





Daily precipitation measurements from the historical Dom Luiz Observatory (IDL) in Lisbon from December 1863 to February 2020 were used allowing for an objective comparison between the different EPE events. Data from 114 automated stations over mainland Portugal was provided by IPMA (Portuguese Institute of Sea and Atmosphere), with a 10-minute temporal resolution. This was used to compute the daily accumulated precipitation between 09 UTC of day 12 December 2022 to 09 UTC of day 13 December 2022, in order to be compatible with the long-term IDL's classical observations since 1863. To extend the assessment of this EPE to the entire Portuguese territory, we employ an inverse distance weighting approach. This method, also known as the inverse distance-based weighted interpolation, uses a weighted mean of nearby observations to estimate unknown values. In this case, values are estimated within the Portuguese territory.

## 2.2 Atmospheric River Detection Method

For the detection of atmospheric river events, we have implemented and adapted the global atmospheric river tracking method developed by Xu et al. (2020) and widely used nowadays (e.g., Fernández-Alvarez et al 2023, Khouakhi et al 2022). The detection algorithm is based on thresholds at the spatiotemporal scale of ARs and is independent of the IVT thresholds.

This method searches for ARs on an anomaly field that is calculated at each time-step. For each IVT field, the algorithm replaces each grid-point with the minimum value that surrounds it, i.e., it searches for the lowest value within the 8 grid-points around it. This 'reconstructed' field is then subtracted to the original IVT, and the result (the anomaly) is used to detect AR objects. Afterwards, a threshold of 300 kg m$^{-1}$ s$^{-1}$ is applied to the anomaly field, and to each region the standard criteria for AR detection (length ≥ 2000 km; width < 1000 km; length/width ration > 2) are implemented. More details on this method can be found in Xu et al. (2020). Additionally, in order to exclude tropical cyclone features from the analysis, we apply both a circularity criteria with a maximum value of 0.5 (following the work of Mahto et al., 2023) and a minimum value of 1000 km to the distance between the first and last points of the AR axis.

## 3 The extreme precipitation's spatial distribution on 13 December

The month of December 2022 was classified by IPMA as a very wet month in terms of precipitation. According to their monthly summary report (IPMA, 2023), the total monthly average precipitation in Portugal has a value of 250.4 mm (corresponding to a 174% of the mean value). This was the second highest value since 2000 (311.5 mm in December 2000). In Figure 1 we present the station-based precipitation measurements. At the IDL station, classical in situ precipitation measurements are available from December 1863 to February 2020. Daily precipitation records obtained at classical stations for any given day n correspond to the precipitation registered between 09 UTC of day n-1 and 09 UTC of day n. In Figure 1a, in a dark red color we present the top five peak values measured until 2020. These include the four events already mentioned in the Introduction section and also the November 1983 episode, where a value of 95.6 mm was measured. Both the 95$^{th}$ (horizontal green line) and the 99$^{th}$ (horizontal purple line) percentiles were added, representing a value of 89.93 and



106.09 mm, respectively. All these top five events are above the 95[th] percentile, but only December 1876, February 2008, and December 2022 are above the 99[th] percentile, confirming the exceptional rainfall events that took place on these days.

The rainfall amounts at IDL between 6 and 20 December 2022 is shown in Figure 1b. The periods 1-5 and 23-31 December
show zero rainfall, and thus they were omitted. The intense rainfall had already struck the Lisbon district on 8 December with a value of 83.3 mm (IPMA, 2023). In addition to this earlier event, three other moderate events occurred with rainfall values ranging between 30 and 40 mm, namely during 11, 14, and 20 December. However, all these moderate peaks were dwarfed by the rainfall value of 120.3 mm measured on 13 December, surpassing all the previous records presented in Figure 1a. In Figure 1c the spatial distribution of rainfall measurements through the Portuguese territory on 13 December is
shown. Rainfall values above 60 mm were measured in Lisbon and the two closest districts. Most of the northern sector of Portugal experienced rainfall values between 15 and 45 mm, with a small region experiencing a rainfall amount higher than 75 mm.

According to the IPMA report (IPMA, 2023), one station (Penhas Douradas – Guarda) featured a rainfall amount of 99 mm but it did not surpass previous records. In addition to the IDL station, several other meteorological stations registered the
165 highest daily precipitation value to date, namely Barreiro (83.4 mm, surpassing 68.3 mm in December 1978), Almada (81.9 mm, surpassing 73.7 mm in January 2004), and Mora (98.8 mm, surpassing 81.5 mm in November 1983). Despite the absence of an official report of this event, flood damage and severe weather led to incidents in several parts of Portugal and Spain, where various Spanish areas registered precipitation values of at least 90 mm (Davies, 2022; AEMET, 2022).

## 4 Pre-Conditions for the Atmospheric River Development

The main characteristics of the large-scale atmospheric circulation, boundary conditions and synoptic features leading to the extreme precipitation event on 13 December 2022 are now analyzed from 5 December onwards. The mean SLP anomaly, 250 hPa wind, 500 hPa geopotential height and temperature from 5 to 11 December are presented in Figure 2. A similar assessment was undertaken for the SST anomalies but for a slightly longer period (1 to 11 December). The mean SLP anomaly respective to the reference to the 1980-2022 period and the 250 hPa wind (contour lines) are shown in Figure 2a.
Wind values higher than 30, 40, and 50 m/s are represented in green, light green, and yellow, respectively. During this period, the upper-level jet stream was located at a southerly location in the North Atlantic, with a direct trajectory towards the western coast of Iberia. The mean SLP anomaly shows a southwest-northeast dipole, with negative anomalies above western Europe and northeast Atlantic, and positive anomalies above Greenland, in line with a negative NAO phase.


Figure 1: Station-based precipitation measurements. (a) Yearly maximum Dom Luiz Observatory's precipitation measurements from 1863 to 2020. The green (purple) horizontal line represents the 95th (99th) percentile. Dark red bars represent the top 5 measurements. In addition, the 2022 record was also added. (b) Precipitation measurements between the days 6 and 22 December at IDL. (c) Precipitation (in mm) from 114 Portuguese stations on 13 December (values calculated between day 12, 09 UTC and day 13, 09 UTC). The black dots show the location of the stations considered.

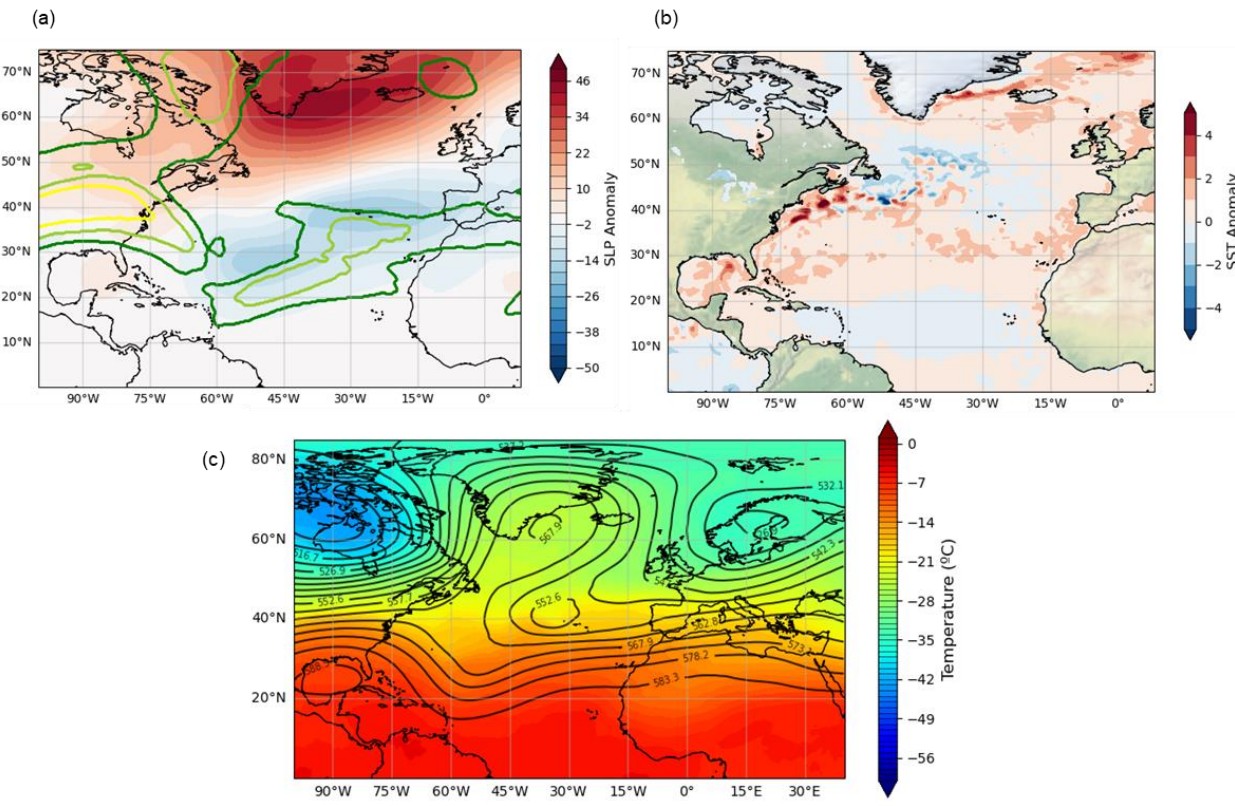

**Figure 2: (a) Mean SLP anomaly field (with reference to the 1980-2022 period) and mean 250 hPa wind (contour lines) (≥ 30 m/s – green; ≥ 40 m/s – light green; ≥ 50 m/s - yellow). (b) Mean SST anomaly field (with reference to the 1980-2022 period). (c) Mean 500 hPa geopotential height (black contours) and temperature (colored region). The SST anomaly is calculated for the period 1 to 11 December and the other variables between 5 to 11 December.**

The mean SST anomalies for 1 to 11 December 2022 compared to the reference period of 1980-2022 are shown in Figure 2b. In the latitudinal band between 20 and 40 ºN, the SST anomalies are positive, with coastal areas of north Africa and Iberia presenting an anomaly value of 1-2 ºC. Additionally, the Gulf Stream presents anomalies above 2 ºC. According to Climate Reanalyzer[2], SST anomalies for the entire North Atlantic were above two times the standard deviation, with a mean SST value of 21.5 ºC (with reference to the 1982-2011 period). The mean 500 hPa geopotential height and temperature fields from 5 to 11 December are presented in Figure 2c. There is a clear north-south dipole with temperatures higher than -20 ºC predominantly located below the 40 ºN latitudinal band, and lower above. Following the same distribution, the geopotential height shows a clear Rex Block Pattern (Rex, 1950; Sousa et al., 2020), located in lower temperatures and a zonal flow equatorward, in higher temperatures. The combination of these two patterns led to the passage of several low-pressure systems on a southward track towards Iberia, leading to several rainfall events throughout December.

---

[2] https://climatereanalyzer.org/clim/sst_daily/

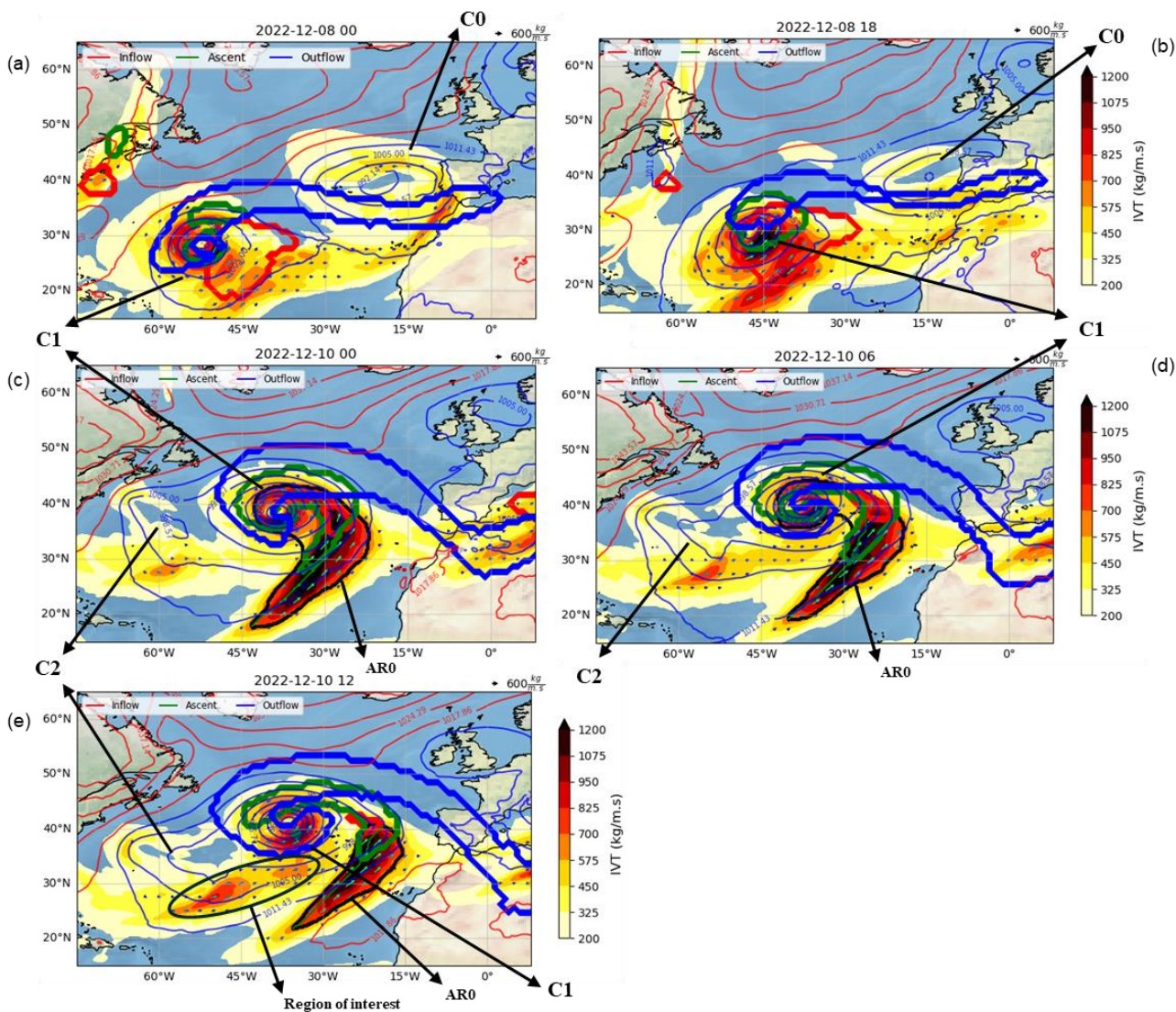

**Figure 3: The spatial pattern of the vertically integrated water vapor transport (IVT) and IVT vectors. The SLP contour lines are represented as blue for pressure below 1013 hPa and red above. The black contour around the highest values of IVT represents the AR spatial distribution and the thin green line inside identifies its axis. Additionally, the WCB data is added, with the bolder contour lines representing inflow (red contour), ascent (green contour), and outflow (blue contour).**

The evolution of the synoptic conditions over the North Atlantic sector on the days prior to the detection of the AR that led to the EPE is presented in Figure 3. Several low-pressure systems were present on the North Atlantic on 6 December, one located near the Azores Island (C0 system) and another close to Porto Rico (C1 system), each connected to a relatively intense IVT. As the C0 system moves towards the Iberian Peninsula, we suggest that the C1 system functioned as a





conveyor, advecting moisture westward towards the area of influence of C0 (Figure 3a). The moisture advection feature extended from the western coast of north Africa to southern Iberia and was associated with intense precipitation through the

215 Portuguese territory on 8 December (83.3 mm, cf. Figure 1b) as reported by IPMA. Later that day, a small low-pressure system (C2 in Figure 3b, 1011 hPa) formed in the northwest Atlantic, with relatively high convergence (divergence) at the low (high) levels (Figure SI2 and Figure SI3, respectively), that contributed to the enhancement of moisture content to its warm sector, probably profiting from the high SST anomalies above the Gulf Stream (Figure 2b) that intensified the evaporation rates. As the C0 system moved eastward over central Europe and dissipated, the C1 cyclone, now located in

central North Atlantic, tracked poleward with substantially higher IVT values, eventually higher than 1000 kg m$^{-1}$s$^{-1}$, leading to the detection of an AR structure (AR0), involving both the region with high IVT and the C1 extratropical cyclone region (Figure 3c). Both cyclones (C1 and C2) merged on 10 December 06 UTC (Figure 3d) and by 12 UTC a secondary IVT maximum could be identified (Figure 3e region of interest, AR1 in Figure 4). This AR was first detected on 10 December 18 UTC (Figure 4a) and is the one associated with the EPE in Iberia on 12-13 December 2022. In the next section, we will

analyze in detail the further development of the ETC and the AR and how they lead to the EPE.

## 5 Large-scale atmospheric conditions associated with the Atmospheric River Life Cycle

The AR algorithm detected a total of four landfalling ARs on western Iberia in December, namely on days 12, 18, 22, and 30 December. The AR that made landfall on 12 December is analyzed in depth here as it was directly responsible for the 24h maximum value ever observed in Lisbon and also lead to a significant precipitation fingerprint on more continental area,

leading to several record-breaking values outside Lisbon. According to our 6-hourly dataset, this AR event lasted for approximately 72 hours with a mean intensity ranging between 688 kg m$^{-1}$ s$^{-1}$ and 848 kg m$^{-1}$ s$^{-1}$ and a maximum IVT ranging between 947 and 1227 kg m$^{-1}$ s$^{-1}$, placing it in a category 4/5 (strong/extreme) AR event based on Ralph et al. (2019) AR category scale, thus making landfall on Portugal as an extreme event. For the sake of simplicity only four time-steps are presented through this section's images, namely the first time the algorithm detected the AR (panels a), the time-step in

which the AR made landfall on the west coast of Iberia (panels b), 24 hours later (panels c), and the last time-step that the method detected the AR (panels d).

The spatial distribution of IVT, and corresponding IVT vectors, the associated configuration of AR, SLP field and WCB phases are presented in Figure 4. The AR was associated with a deep extratropical cyclone (merger of C1 and C2) with a minimum value of approximately 960 hPa on 10 December. A deep ETC leads to intensified cyclonic winds at the low levels

that can intensify the moisture-driven IVT on the AR (e.g., Zhang et al. (2019) and Eiras-Barca et al., 2018). At the AR's initial stage, first detected on 10 December 18 UTC (Figure 4a), a small region of moisture inflow was located at the center of the AR area (red contour), intensifying the object's moisture content. After 36 hours the AR system made landfall on the western coast of Portugal (Figure 4b) on 12 December 06 UTC, with a mean IVT value of 792 kg m$^{-1}$s$^{-1}$ and an inflow area located within the maximum IVT values, meaning that the system was still evolving through the convergence of moisture



within its area, resulting from the WCB's inflow phase. This convergence of moisture along the AR path has been proved essential for maintaining the structure and ensuring the development of the system (Hu and Dominguez 2019). The area located northward of Iberia shows an uplift of moisture and a small outflow region, meaning that the associated dynamic uplift played a major role in achieving the high precipitation (Figure 1c). On 13 December the AR had lost more than half of its spatial extent and it was mostly affecting central and south of Portugal, some regions of Spain and the north of Africa

(Figure 4c), in agreement with precipitation recorded over continental Portugal (Figure 1c). However, its moisture content was still high, leading to intense rainfall on a few regions of Spain and incidents due to severe winds. At this time, there was an inflow of moisture around the southwest coast of Portugal and an ascent region just northward of the AR. This ascent motion fits well the divergent pattern, showing a maximum just outside of central Portugal (Figure 5c), a pattern that is related to the cyclone's mesoscale motion. The last step where the algorithm defined the object as an AR was 13 December

18 UTC (Figure 4d), when the system was mostly affecting southern Spain and north Africa. There was still an inflow of moisture just outside of the west coast of Africa, but there was a large ascent and outflow motion counteracting the inflow, leading to the dissipation of the system.

On the Portuguese coast, orographic uplift is less important e.g. in California (e.g., Lavers and Villarini, 2015). Thus, we analyze the features of the cyclone associated with the AR. With this aim, convergence fields are evaluated at two pressure

levels, namely the convergence at low levels (850 hPa) and the divergence at high levels (200 hPa). Enhanced convergence at low levels co-located with enhanced divergence at upper levels indicated large-scale ascent, and this optimal conditions for cloud and precipitation generating conditions. Additionally, the jet stream and associated fronts are analyzed with help of 250 hPa winds the DWD weather charts, respectively. Figure 5 represents the IWV combined with the convergent pattern at low levels; and Figure 6 represents the winds at 250 hPa combined with the divergent pattern at high levels (left column) and

the meteorological weather charts (right column).

Figure 5 shows the integrated water vapor (IWV) as well as the low-level convergent pattern at 850 hPa. IWV values ranging between 25 and 35 kg/m$^2$ were present within the AR area, with a convergent pattern close to the defined axis. As the system developed, the convergence of moisture intensified the AR and by the time that it made landfall on 12 December, IWV values exceeded 37.5 kg/m$^2$ throughout the AR axis (Figure 5b). The landfall on the west coast of Iberia intensified the

convergent field inside the AR area that was above land (Figure 5c). The system started to decay and on 13 December 18 UTC the moisture content was within 25-30 kg/m$^2$ with convergence along its entire area (Figure 5d). Furthermore, based on the 500 hPa geopotential height field (black contour lines on SI.1), it is recognizable that this AR travelled along a trough, with the ETC located northward and an anticyclone placed to its south side. Based on the study of Guo et al. (2020), this tripole pattern leads to an intensification of the horizontal pressure gradient that can force the transport of moisture to the

AR.

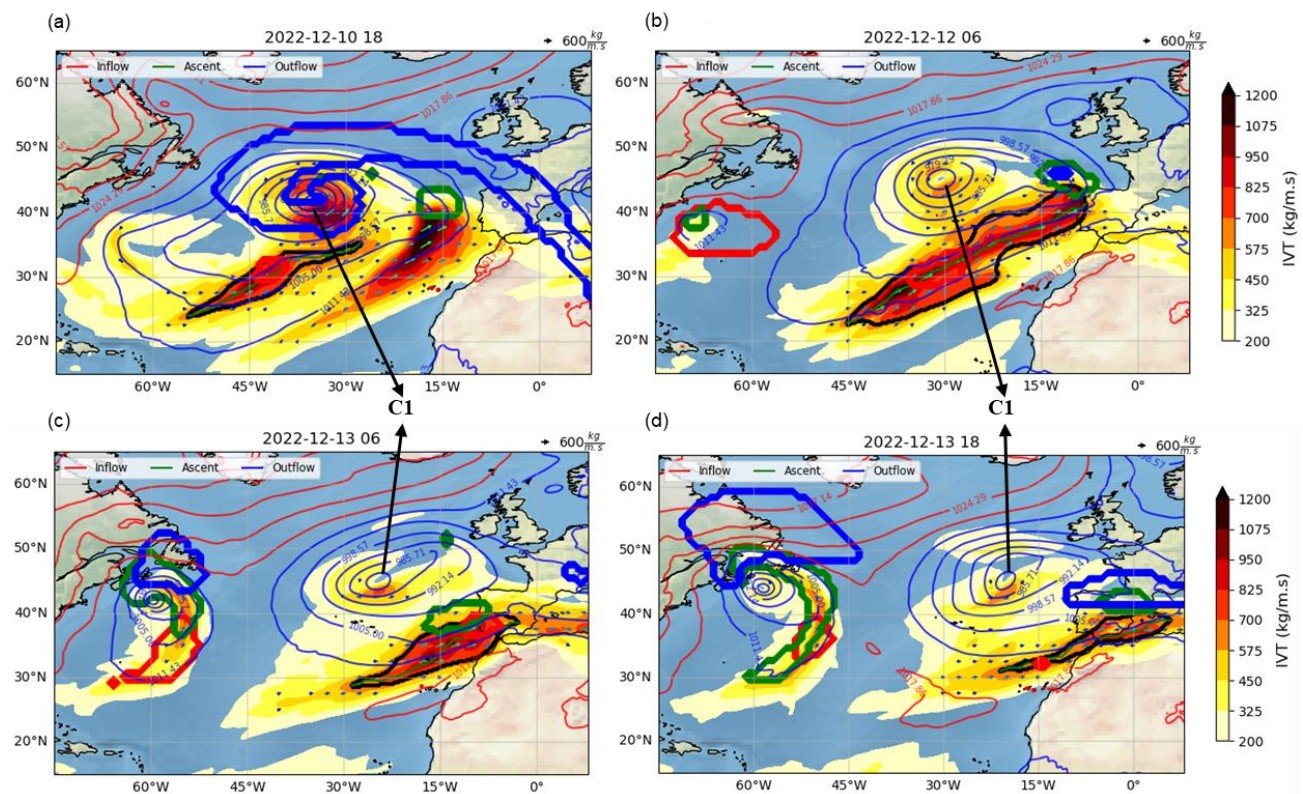

**Figure 4: The spatial pattern of Vertically integrated water vapor transport (IVT, shaded) and IVT vectors. The SLP contour lines are represented in blue for values below 1013 hPa and red above. The black contour around the highest values of IVT represents the AR spatial configuration and the thin green line inside identifies its axis. Additionally, the WCB data is added, with the bolder contour lines representing inflow (red contour), ascent (green contour), and outflow (blue contour).**

In Figure 6 we analyze the development of the jet stream with the divergent pattern at 250 hPa (left column) and the corresponding synoptic DWD weather charts (right column). The west coast of Iberia has been marked with a red contour line on the DWD weather charts for easier identification. The jet stream is located at comparatively low latitudes over the central North Atlantic, in a Southwest – Northeast orientation pointing towards the Iberian Peninsula. From its genesis to landfall, most of the AR area was embedded within upper (250 hPa) wind values of 50 to 70 m/s. However, as the system made landfall (Figure 6c), the AR and the cyclone warm sector got detached from the wind high values, decaying one day after. This detachment is observed on the DWD weather charts (Figure 6 right column), in which the occlusion front associated with the ETC is affecting the AR system. While high wind values are present, by the time a new front is observed within the AR area, associated with a small secondary low-pressure system. A larger region of upper-level divergent flow (blue contours on Figure 6, left column) is observed within the AR area, co-located to the convergent flow at low levels (blue contours on Figure 5). This may have supported the formation of a secondary cyclone off the north coast of Iberia, that led to the warm front to rise and diverge as the cold front wrapped up. As the system approached the European continent, the cold front was present within the AR area. This is true for the region close and within Iberia, meaning that some of the


rainfall that occurred might have been generated by the ascent of moisture on the warm front, but also to the AR area

northward of its axis.

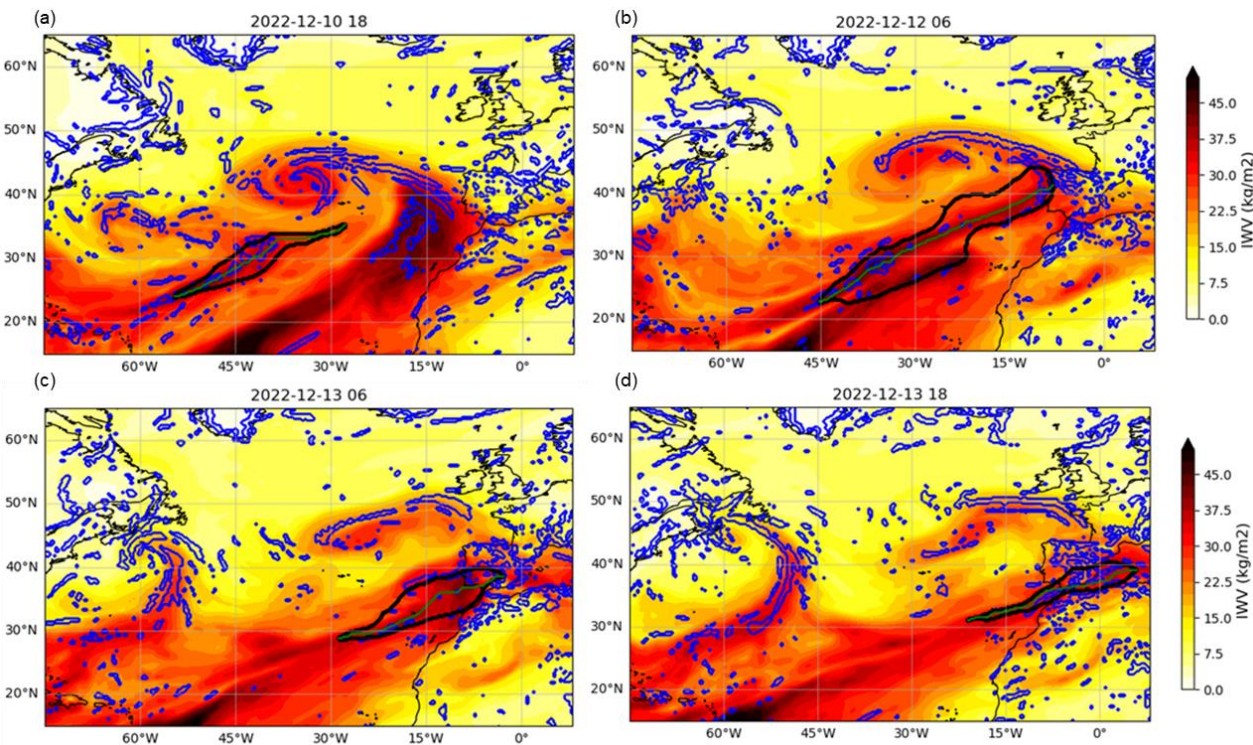

**Figure 5: Spatial pattern of the Integrated Water Vapor (IWV, shaded). The blue contours represent the convergence of the wind**
**field at 850 hPa (only represented for values lower than -0.3x10$^{-4}$ s$^{-1}$). The black contour (thin green line) represents the AR area**
**(axis).**

## 6 Conclusions

The western coast of the Iberian Peninsula is susceptible to the occurrence of extreme rainfall events, resulting in

socioeconomic impacts, such as floodings and landslides (e.g., Liberato and Trigo, 2014; Fernández-Montes et al., 2014). On

12 and 13 December 2022, the western coast of Portugal was affected by an extratropical cyclone and associated

atmospheric river, leading to an extreme precipitation event. This event registered 120.3 mm of precipitation at the IDL

station in Lisbon, surpassing all previous events measured continuously since December 1863. The previous maximum daily

rainfall at IDL in Lisbon was registered on 18 February 2008 - 118.4 mm. The objective of this study was to analyze the

synoptic features associated with the occurrence the all-time 24h record-breaking precipitation values recorded in Lisbon,

Portugal between the 12 and 13 December 2022, in particular the role of the involved AR and ETC.

**Figure 6: Spatial pattern of the 250 hPa wind speed (right column, shaded), with the blue contours representing the high-level divergent pattern for values higher than 0.3x10⁻⁴ s⁻¹. C1 system center is marked in each subplot to facilitate visualization comparison with previous figures. The right column represents the DWD synoptic weather charts, retrieved from the wetter3 archive. The west coast of the Iberian Peninsula is marked with a red contour.**



The first research question dealt with the pre-conditions associated with the ETC and AR formation. The jet stream was located around 30ºN, resembling the southern jet stream location in following Woollings et al (2010), with a clear trajectory towards western Iberia, and corresponding to a negative NAO phase. According to Ramos et al. (2015), this large-scale circulation pattern increases the probability of AR occurrence in southern Europe. Additionally, December 2022 was characterized with above normal positive SST anomalies which could allow to potentially have a higher evaporation from the north Atlantic Ocean. In addition, the geopotential height at mid-levels shows a zonal flow at low latitudes, following the jet stream pattern, and a clear Rex Blocking pattern at high latitudes near Greenland.

The combination of these synoptic features resulted in the development of several low-pressure systems at comparatively low latitudes over the Atlantic Basin, all travelling along the same path towards western Europe. Specifically, to western Iberia, the combination of the dominant westerly flow and embedded cyclones lead to multiple intense precipitation events through the month of December (Figure 1b). As stated by Dacre et al. (2015), a secondary cyclone that follows in the footprint of a previous cyclone may profit from the high moisture content left behind by the first one, potentially leading to more intense precipitation in subsequent cyclones. This sequence of events can be appreciated in our results where we observe that the C1 system, to which the AR was associated following the C0 system's path. In this context, we hypothesize that the moisture left behind by the first ETC was gathered by the follow up C1 cyclone (Fig. 3).

The AR was first detected by the algorithm on 10 December, 18 UTC. Results show that this object was formed by the merge of two cyclones: (1) an ETC already present on the North Atlantic, characterized with a high moisture content; and (2) a small low-pressure system formed in the northeast coast of North America, that was associated with enhanced moisture uptake as it moved over the Gulf Stream, a region characterized by an above normal positive SST anomaly (Figure 2). Moisture from the two cyclones, in combination with the high convergence pattern in central North Atlantic, allowed the moisture content increase culminating with the AR formation.

Regarding the second research question, related to what mechanisms can be associated with the AR movement and rainfall occurrence, as it made landfall on the western coast of the Iberian Peninsula. It was found that throughout its entire life cycle an inflow of moisture occurred within the AR's area, associated with the inflow of the WCB of the ETC. This allowed the AR to evolve and maintain its characteristics for 72 hours, most of the time as an extreme AR event. On 12 and 13 December, the WCB ascent was located within the western coast of Iberia, connected to an occlusion formed off the north-western coast. Combined with a high pattern of both convergence at low-levels and divergence at upper-levels, the system was able to generate intensified rainfall affecting almost the entire Portuguese territory but affecting central Portugal in an extremer way.

The interpretation of all variables allows us to conclude that this high-impact event was associated with: (1) an intense ETC that supported the formation of an intense AR, combined with favorable strong convection and fueled by high moisture content (due to the presence of the WCB's inflow located within the AR area throughout its life cycle) in the region associated with the ETC; (2) a low latitudinal location of the jet stream with a clear path towards western Iberia; (3) large-scale ascent due to the combine effect of high convergence at low levels and divergence at high levels; and (4) the presence





of a front that could have intensified the convective motion locally over the Iberian Peninsula in addition to the WCB's ascent located around the same mean location.

These results are also important in the scope of ongoing climate change. The increase in global mean surface temperature in the last decades inevitable lead to an increase on the water-holding capacity of the atmosphere and generally to more extreme precipitation events (IPCC, 2021). The enhanced number of extreme precipitation events in the 160-year IDL time series (see Figure 1a) in the last two decades, with extreme values in 2004, 2008, 2013 and 2022 (this study) is consistent with this assumption. Regarding AR activity affecting the Iberian Peninsula, Ramos et al. (2016), provided evidence of an increase in ARs frequency in future climates by the end of the twenty-first century. However, this increase is lower and more uncertain for Iberia than other Atlantic regions in Europe. Although the increase of AR occurrence is directly linked to the increase in their moisture content, a robust precipitation decrease over Iberia is identified in the last decades of the twenty-first century (Sousa et al., 2020). This indicates a possible extension of stable and dry summer conditions (Hadley Cell expansion) and a decoupling between moisture availability and dynamical forcing for the region. This indicates that further research is needed towards a better understanding of extreme precipitation events over Iberia, the role of atmospheric rivers and the associated synoptic forcing in a warming climate.

## Acknowledgements

This work was supported by the Portuguese Science Foundation (FCT) I.P./MCTES through the project AMOTHEC (DRI/India/0098/2020) with DOI 10.54499/DRI/India/0098/2020 (https://doi.org/10.54499/DRI/India/0098/2020) and through national funds (PIDDAC) – UIDB/50019/2020 (https://doi.org/10.54499/UIDB/50019/2020), UIDP/50019/2020 (https://doi.org/10.54499/UIDP/50019/2020) and LA/P/0068/2020 (https://doi.org/10.54499/LA/P/0068/2020). TMF was supported by FCT through PhD grant UI/BD/154496/2022. JGP thanks AXA Research Fund for support. AMR was supported by the Helmholtz "Changing Earth – Sustaining our Future" program.

## Data Availability

ERA5 reanalyses are available from the Copernicus Climate Data Store (https://doi.org/10.24381/CDS.ADBB2D47, Copernicus Climate Change Service, 2023). The Lisbon's Dom Luiz Observatory precipitation data were provided by Dr. Maria Antónia Valente and are available at http://dados-met-idl.campus.ciencias.ulisboa.pt/precipitation/. Precipitation data from IPMA is available at https://www.ipma.pt/pt/produtoseservicos/index.jsp?page=dataset.pt02.xml (last access: December 2023, Belo-Pereira et al., 2011). The DWD surface maps are freely available through the following archive (https://wetter3.de/archiv_dwd_dt.html). The WCB data was acessed through the LSDF Online Storage service of KIT, provided by Quinting and Grams (2022).



**Author Contributions**

AMR, RMT and TMF conceived and designed the study. TMF performed the data analysis and prepared the figures. AMR, RMT, THG and JGP contributed with methodologies. The initial draft was written by TMF, supported by RMT, AMR and JGP. All authors contributed to discussions, comments, and text revisions.

**Competing Interests**

One of the authors (JGP) is member of the editorial board of Natural Hazards and Earth System Sciences.

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
