# Peer review of "The Record-Breaking Precipitation Event of December 2022 in Portugal"

_Natural Hazards and Earth System Sciences, 2024_

## Author Comment (AC1)

We thank Reviewer 1 for his/her comments and suggestions, which helped to improve the manuscript and to remove ambiguities/misunderstandings. Below we provide point-by-point responses to each comment.

**Minor changes:**

Line 61: Change "sup-polar"
Line 84: Change "to analyze of the spatial distribution" to "to analyze the spatial distribution"

**Answer:** We thank the reviewer for suggesting these improvements. We will change accordingly in the revised manuscript.

Figure 3: Improve readability. I cannot see "the thin green line inside…". Additionally, it is not necessary to place a black contour around the highest IVT values. If retaining them, briefly explain what C1, C2 and AR0 are in the caption. What does the blue color over the ocean mean? If possible, live open ocean white to improve figure clarity.

**Answer:** The AR axis (thin green line) contour will be expanded in the revised manuscript and, in addition the inflow/ascent phase of the warm conveyor belt will be changed from red/green to a light blue/magenta color to enhance clarity. The black contour around the highest IVT values represents the detected AR by our methodology. As the clear representation of this AR represents one of the main focuses of the paper, we opted to maintain the black contour for the AR.

The meaning of acronyms C1, C2, and AR0 are explained throughout the text, but we agree that they need to be explained in the figure caption. C1 is the main extratropical cyclone, associated with the AR that led to the precipitation record; C2 is a small secondary low-pressure system that transported moisture from the Gulf Stream towards central North Atlantic before merging with C1 and developing the AR; and AR0 is an Atmospheric River (AR) that occurred prior to the main AR but lost its AR characteristics before making landfall. This will be added to the figure caption.

The thin blue/red contours over the ocean represent the SLP data below/above 1013 hPa, that will be changed to black in the revised version removing the difference between high and low pressures (see below the new version of the Figure).

We agree with the reviewer that there's no need to include the blue ocean color and therefore, this will be removed in the revised version of this figure. An updated version of Figure R1 below).

[Figure]

Figure R1 - Spatial patterns of the vertically integrated water vapor transport (IVT, shaded colors) and IVT vectors. The corresponding SLP fields are represented with black contour lines. The AR limits are represented with the bold black contour around the highest values of IVT while the green line inside identifies its axis. Additionally, the WCB data is added, with the bolder contour lines representing inflow (light blue contour), ascent (pink contour), and outflow (blue contour). C1 is the main extratropical cyclone, associated with the AR that led to the precipitation record; C2 is a small secondary low-pressure system that transported moisture from the Gulf Stream towards central North Atlantic before merging with C1 and developing the AR; and AR0 is an AR that occurred prior to the main AR but lost its AR characteristics before making landfall.

Lines 234-236: I assume that panels description refer to those in figures 4 and 5, please carefully state this in text if correct.

**Answer:** Yes, the explanation of showing just 4 time-steps is extensive to Figures 4, 5 and 6. It should be noted that on Figure 6 we are representing two different patterns in each column, meaning that the 4 time-steps representing 250 hPa wind speed and high level divergence pattern corresponds to figure 6a, 6c, 6e, and 6g (left panels) and the DWD weather charts correspond to figure 6b, 6d, 6f, and 6h (right panels). This information will be explicitly mentioned in the new version of the manuscript.

Figure 4: Please improve clarity as in figure 3. I cannot see the red contour ("small inflow area") in panel 4a, maybe masked by the IVT shaded area. Had to go big zoom to find something resembling a red contour. Can not see the inflow area (red contour) in the maximum IVT area

**Answer:** The answer to this issue is related to a previous answer. As explained when answering Figure 3's question raised above, we will change in the revised version of the manuscript the color of the WCB inflow from red to light blue and the color of the ascent phase, as stated before, from green to magenta. The result can be found below in Fig R2.

[Figure]

Figure R2 - Spatial patterns of the vertically integrated water vapor transport (IVT, shaded colors) and IVT vectors. The corresponding SLP fields are represented with black contour lines. The AR limits are represented with the bold black contour around the highest values of IVT while the green line inside identifies its axis. Additionally, the WCB data is added, with the bolder contour lines representing inflow (light blue contour), ascent (pink contour), and outflow (blue contour). C1 is the main extratropical cyclone, associated with the AR that led to the precipitation record.

Lines 329-330: Change "This sequence of events can be appreciated in our results where we observe that the C1 system, to which the AR was associated following the C0 system's path." I cannot understand the meaning of this sentence.

**Answer:** We agree with the reviewer that the above-mentioned sentence could be misleading. In Dacre et al 2015, the authors stated that a secondary cyclone that follows the same mean path of a previous cyclone may incorporate moisture left behind by the first one. Here we assume that, since the two cyclones followed the same overall path through the North Atlantic, the second cyclone may have made use of the moisture available on the atmosphere to intensify. This hypothesis is supported by the intensification and increase of spatial extent of the AR and the convergence pattern through its life cycle. We add this information to the manuscript in order to make our statement understandable.

Lines 338-339: Rewrite, the sentence does not seem complete

**Answer:** We thank the reviewer for the comment. We agree that there was a lack of connection between the two phrases. In the revised manuscript we will point to the relationship between the warm conveyor belt inflow phase and the intensification of the AR moisture content and its area. Therefore, the sentence will be changed to: 'Regarding the second research question, related to what mechanisms can be associated with the AR movement and rainfall occurrence, as it made landfall on the western coast of the Iberian Peninsula, it was found that throughout its entire life cycle an inflow of moisture occurred within the AR's area, associated with the inflow of the WCB of the ETC.'

---

## Author Comment (AC2)

We thank Reviewer 2 for his/her comments and suggestions, which helped to improve the manuscript and to remove ambiguities/misunderstandings. Below we provide point-by-point responses to each comment.

**Specific Comments**

Section 1 – could you mention how many ARs affect Portugal each year? (line 64) and how much precipitation input they provide? Can you also define a little better what you mean by "infer the evolution of rainfall" – like trends throughout the period? Maybe mention this? Also, why give the top three rainfall events from 1863 to 2013 instead of in the whole record? What are the incidents reported by the Portuguese Civil Authority – are they rescues, or damage reports, or both? I think maybe some more clarity and context in this part of the Introduction would benefit the study. I suggest also when you define the goals for the paper in the last paragraph of the section, you could remind the reader again why this case study is important to understand – because it was so impactful, of course – but could it be better predicted if it was better understood? Are we expecting more events of this nature more frequently? Any other relevant literature on frequency of extremes or information on the predictability that could be included in Section 1?

**Answer:** According to Ramos et al 2015, the top 6% most intense precipitation events in the Lisbon area occur in conjunction with an AR lasting longer than 2 time-steps, with a mean monthly occurrence of 2 ARs per winter month (October-March) from 1979 to 2012. In addition, Hay and Williams 2023 state that ARs deliver 20-30% of annual precipitation in western Europe and the western US.

There is not a clear trend in the IDL station measurements. The mean of 'infer the evolution of rainfall' resides on the fact that the 160-year long record allows us to see the interannual variation of precipitation, and the 'return period' of intense precipitation events. For example, we show the top 10 events of the station (Table 1). With the exception of the 1876 event, all the other 9 top events occurred within the last 60 years.

| Time of Occurrence | Precipitation Measurement (mm) |
|---|---|
| 13-12-2022 | 120.3 |
| 18-02-2008 | 118.4 |
| 05-12-1876 | 110.7 |
| 24-10-2013 | 102 |
| 30-01-2004 | 101.2 |
| 19-11-1983 | 95.6 |
| 19-10-1997 | 92.6 |
| 02-11-1997 | 91.3 |
| 11-10-1962 | 91.2 |
| 07-12-2012 | 91 |

We will rephrase since those records are applicable to the entire record.

Reports from the Portuguese Civil Authority on the December 2022 event include flooding in several downtown areas, a number of buildings including a hospital were damaged, several road and vehicular tunnels were closed, and parts of the airport were also flooded. We believe that it could be better predicted if it was better understood. With that in mind, we made both a thermodynamical and dynamical analysis of the AR and atmospheric circulation around.

From a climate change perspective several works found a clear increase in the IVT maximum distribution and an increase in both intensity and frequency of the ARs in the future under different warming scenarios (e.g. Ramos et al 2016).

There is more relevant literature on frequency of extremes but related to other regions. Here we tried to include only those related to cases on the Iberian Peninsula. Concerning the predictability of ARs on the Iberian Peninsula, e.g., Ramos et al 2020 have analyzed the predictability of precipitation considering (or not) the role of ARs and found that for short lead times the direct forecast of precipitation is more accurate than indirect precipitation forecast based on IVT. However, these authors also show evidence that for longer lead times (around the 10th day), there is an added value to predict rainfall in western Iberia considering the associated forecast of IVT.

Line 111- I don't think this sentence is necessary – I'm not sure how you would justify not including the whole column or where you would stop computation.

**Answer:** We apologize for the misunderstanding in the text. Since ERA5 has already available a vertical integral field of the northward and eastward water vapor flux, using the entire atmospheric column, it removes the additional computation step as it is computed internally by the ERA5. In the revised manuscript we have changed the text to clarify this issue.

Section 2.1 – I am not clear on why not use the entire IDL daily precipitation record.

**Answer:** We thank the reviewer for the comment. The classical measurements at the IDL station (performed by an observer) are available from December 1863 (the beginning of the station) until March 2020. However, due to covid-19, only the automatic station's measurements are available since March 2020.

Section 2.2 – Is there any additional justification on the detection methodology other than its wide use? What about it makes it so widely applicable and in particular applicable here?

**Answer:** Ramos et al. 2015 developed an AR detection tool that allowed the identification of ARs affecting the Iberian Peninsula. This method applies the local 85$^{th}$ percentile of IVT distribution and standard geometrical criteria in order to identify the events. However, this method only detects ARs when the object makes landfall on the western coast of Iberia, i.e., at the final stages of the AR life cycle. With that in mind, a global AR detection algorithm was applied to the study of this extreme precipitation event in order to better understand the mechanisms behind its occurrence. In addition, this method (Xu et al. 2020) is available online as a free source code.

Section 3 – what is the main takeaway from the spatial distribution? Is it the widespread nature of the event? There is also some discussion here about the record-breaking nature and percentile value associated with the given events. Consider keeping the discussion narrower and focused on the spatial distribution and what we are meant to take away from it, or perhaps provide all the same information (in a table?) for different stations that have had their records broken and include the percentile information, so we are getting the spatial sense of that as well.

**Answer:** Yes, the main takeaway from the spatial distribution maps of the daily precipitation is to emphasize the widespread nature of the event. As we can see in the Figure 1 almost the entire Portuguese territory was affected by the AR, but only the central part and one station in the central-northern region registered precipitation values higher than 90 mm. The discussion about the record-breaking nature and percentile values was just to place the event in a broader climatological context of this historical station. The addition of the other 4 record-breaking stations was mainly to emphasize how extreme the event was also outside Lisbon.

Line 170 – add reason behind starting to evaluate the synoptic conditions on 5 December and SSTs on 1 December.

**Answer:** The different time scales reside on the fact that the SST changes on a longer time frame than the atmospheric conditions. This issue will be made clearer in the revised version of the manuscript.

Figure 2- add units for a/b color bars.

**Answer:** We thank the reviewer for the comment. We agree there's a lack of units on both anomalies. We will add 'hPa' to the figure a (SLP anomaly) and '°C' to the figure b (SST anomaly) in the revised paper.

Section 4 – are the noted patterns e.g. in Figure 2 usual patterns for extreme events? Or what other meaning should the reader take from the evaluation?

**Answer:** The main objective of this figure is to show that suitable large-scale conditions were available for the formation of both the extratropical cyclone and the associated atmospheric river. These conditions include a higher than usual SST in the areas where the AR will obtain additional moisture (Gimeno et al 2020), the clear Rex blocking structure (Trigo et al. 2004) that allowed the formation of several low pressure systems, the low latitude location of the jet stream (Woollings et al 2010) with a clear trajectory from central north Atlantic towards Iberia, and a NAO negative phase that has been showed to intensify the AR occurrence in southern Europe (Ramos et al 2018).

Line 212- what is considered relatively intense IVT? Over 500 units? Consider adding this information. Also – should "suggest" be "hypothesize"? (similar note to "probably" in Line 218). What would be needed to test these hypotheses (even if not in scope of this paper)?

**Answer:** Yes, we consider relatively intense IVT values over 500 $kg.m^{-1}.s^{-1}$, we will add this information to the revised version of the manuscript. Regarding the second part of the comment, we believe that, in particular, a water vapor trajectory analysis would allow us to confirm both hypothesis: (1) the moisture transport through C1 towards C0; and 2) the inflow of moisture within C2 above the Gulf Stream, due to high positive SST anomalies.

Line 242 – should "intensifying" instead be something like "supplying"?

**Answer:** We believe that either is applicable since the objective is to tell the reader that the AR area increased from the inflow of moisture. This inflow was due to the presence of the inflow phase of the warm conveyor belt, verified with the convergent pattern at low levels in Figure 5.

Line 250 – should this be that the IVT as a whole is still high in terms of both moisture and winds?

**Answer:** Yes, this phrase is related to the IVT as a whole (both moisture and winds) that is later confirmed by the integrated water vapor content and also the dynamical component (ratio IVT/IWV, supplementary information, Figure SI.1). Moisture contents show values higher than 25 kg/m2 and the dynamical component include wind values higher than 20 m/s.

Line 258 – I think the first e.g. should be something like "than in locations like California with significant orography at the coast". The authors are stating that dynamic forcing is more important right? Perhaps state that explicitly and then note that you will evaluate that forcing.

**Answer:** We thank the reviewer for this comment. We agree that we should clarify and explicitly state both ideas in the revised manuscript.

Figure 6 – could the same map view be used for both columns?

**Answer:** The 250 hPa wind extent is useful in order to see the jet stream location along the North Atlantic. However, the DWD weather charts are retrieved directly from their database site which doesn't allow us to adjust the spatial extent.

Line 289-290 – I think this is an incomplete sentence.

**Answer:** We thank the reviewer for this comment. The correct phrase should read: 'While high wind values are present, by the time a new front is observed within the AR area, associated with a small secondary low-pressure system, a larger region of upper-level divergent flow 290 (blue contours on Figure 6, left column) is observed within the AR area, co-located to the convergent flow at low levels (blue contours on Figure 5).'

Section 6 – similar comments to the introduction – I think placing these results in the context of what we can do with them is important. The conclusion here is more of a summary, which is fine, but I think it would benefit from more discussion on implications and utility of the results. Some of this is in the last paragraph but I think it should be highlighted throughout the paper, along with information about what is needed to test the hypotheses mentioned in the paper.

**Answer:** We thank the reviewer for this comment. Due to the increased importance of ARs for both freshwater resource and impacts, a lot of effort has been placed in the analysis and connection between these systems and the large-scale atmospheric circulation to which they are connected, e.g. extratropical cyclones. The prediction of these systems presents variations, namely on landfall location or intensity values. Given the fact that the large-scale circulation is more accurately represented in forecasts than moisture-related fields, a better characterization between ARs and the surrounding systems could be an effective way to improve the prediction skill of ARs (DeHaan et al. (2023); Lavers et al. (2020)). In this paper, we analyze the synoptic background associated with an extreme precipitation event and found several patterns that might have influenced the AR formation and the precipitation generation through the Portuguese territory, leading to the record-breaking precipitation values. These include (1) the occurrence of several extratropical cyclones; (2) high atmospheric moisture contents due to high positive SST anomalies; and (3) the presence of the inflow and ascent phases of the warm conveyor belt that might have helped to intensify the AR's moisture content and precipitation formation. This complementary information will be made clearer in the revised version of the manuscript.

Line 338-340 – This sounds like an incomplete sentence, please rephrase.

**Answer:** Thank you for the comment. There was a lack of connection between the two phrases. Correct phrase should read: 'Regarding the second research question, related to what mechanisms can be associated with the AR movement and rainfall occurrence, as it made landfall on the western coast of the Iberian Peninsula, it was found that throughout its entire life cycle an inflow of moisture occurred within the AR's area, associated with the inflow of the WCB of the ETC.'

Line 344 – rephrase "but affecting central Portugal in an extremer way" to something like "with the most extreme impacts/rainfall in central Portugal"

**Answer:** We thank the reviewer for this comment. We agree with the need to change this sentence and will modify accordingly it in the revised manuscript.

**Technical Corrections**

A general quick read for grammar/word choice (clarity)/readability is warranted although generally the paper is in good shape. A few suggested changes are below (non-exhaustive).

Line 15 – remove "an" before "above normal"

Line 45 – I think you mean to remove "may lead to EPEs" given the rest of the sentence

Line 56 – should be "resulting in" instead of "resulting on"

Line 62 – I think you can remove "multiple" as it is redundant

Line 74 – introduce what is "IPMA" here instead of later in the paper

Line 75 – I think you could put this footnote instead on Line 66 when noting the daily data record.

Line 132 – consider changing "nowadays" to in recent years

Lines 225/229 – "lead" should be "led"

Line 354 – should be "inevitably"

**Answer:** We thank the reviewer for suggesting these improvements. We will change accordingly in the revised manuscript.

**REFERENCES**

Dacre, H. F., Clark, P. A., Martinez-Alvarado, O., Stringer, M. A., & Lavers, D. A. (2015). How do atmospheric rivers form? Bulletin of the American Meteorological Society, 96(8), 1243–1255. https://doi.org/10.1175/BAMS-D-14-00031.1

DeHaan, L. L., Wilson, A. M., Kawzenuk, B., Zheng, M., Monache, L. D., Wu, X., Lavers, D. A., Ingleby, B., Tallapragada, V., Pappenberger, F., & Ralph, F. M. (2023). Impacts of Dropsonde Observations on Forecasts of Atmospheric Rivers and Associated Precipitation in the NCEP GFS and ECMWF IFS Models. *Weather and Forecasting*, *38*(12), 2397-2413. https://doi.org/10.1175/WAF-D-23-0025.1

Gimeno, Luis & Vazquez, Marta & Eiras-Barca, Jorge & Sorí, Rogert & Stojanovic, Milica & Algarra, Iago & Nieto, Raquel & Ramos, Alexandre & Durán-Quesada, Ana & Dominguez, Francina. (2019). Recent progress on the sources of continental precipitation as revealed by moisture transport analysis. Earth-Science Reviews. 201. 103070. 10.1016/j.earscirev.2019.103070.

Lavers, D.A., Ralph, F.M., Richardson, D.S. *et al.* Improved forecasts of atmospheric rivers through systematic reconnaissance, better modelling, and insights on conversion of rain to flooding. *Commun Earth Environ* **1**, 39 (2020). https://doi.org/10.1038/s43247-020-00042-1

Ramos, A. M., Trigo, R. M., Liberato, M. L. R., & Tomé, R. (2015). Daily Precipitation Extreme Events in the Iberian Peninsula and Its Association with Atmospheric Rivers. https://doi.org/10.1175/JHM-D-14-0103.1

Ramos, A.M.; Trigo, R.M.; Tomé, R.; Liberato, M.L.R. Impacts of Atmospheric Rivers in Extreme Precipitation on the European Macaronesian Islands. *Atmosphere* **2018**, *9*, 325. https://doi.org/10.3390/atmos9080325

Ramos, Alexandre & Sousa, Pedro M & Dutra, Emanuel & Trigo, Ricardo. (2020). Predictive skill for atmospheric rivers in the western Iberian Peninsula. Natural Hazards and Earth System Sciences. 20. 877-888. 10.5194/nhess-20-877-2020.

Trigo, R.M., Trigo, I.F., DaCamara, C.C. *et al.* Climate impact of the European winter blocking episodes from the NCEP/NCAR Reanalyses. *Climate Dynamics* **23**, 17–28 (2004). https://doi.org/10.1007/s00382-004-0410-4

Woollings, T., Hannachi, A. and Hoskins, B. (2010), Variability of the North Atlantic eddy-driven jet stream. Q.J.R. Meteorol. Soc., 136: 856-868. https://doi.org/10.1002/qj.625